# Renal Biopsy Diagnosis of Acute Tubular Injury after Pfizer-BioNTech COVID-19 Vaccination: A Case Report

**DOI:** 10.3390/vaccines11020464

**Published:** 2023-02-17

**Authors:** Yu Soma, Daiyu Kitaji, Kaoru Hoshino, Sumire Sunohara, Takehisa Iwano, Naomi Kawano

**Affiliations:** 1Department of Nephrology, Yokohama Minami Kyousai Hospital, 1-21-1 Rokuura-higashi, Kanazawa-ku, Yokohama 236-0037, Japan; 2Department of Pathology, Yokohama Minami Kyousai Hospital, 1-21-1 Rokuura-higashi, Kanazawa-ku, Yokohama 236-0037, Japan

**Keywords:** COVID-19 vaccination, acute kidney injury, acute tubular injury, renal biopsy

## Abstract

Coronavirus disease 2019 (COVID-19) is a severe respiratory infection that can be fatal in unvaccinated individuals; however, acute kidney injury (AKI) is a rare adverse reaction to COVID-19 vaccination. AKI resulting from multiple conditions can have severe consequences, including end-stage renal failure, if not treated with immunosuppressive agents. However, acute tubular injury (ATI) as the sole cause of AKI has not been previously reported. Herein, we discuss an obese 54-year-old man with type 2 diabetes who received four COVID-19 vaccines; three from Pfizer and one from Moderna. Diabetic retinopathy, urinary protein, and occult blood were absent with no other underlying diseases. There was no history of COVID-19 infection. He was referred to our hospital 5 days after receiving the fourth Pfizer-BioNTech COVID-19 vaccine dose with stage 3 AKI. Urinary findings revealed new proteinuria and glomerular occult blood. Physical examination and infection testing were unremarkable. Steroids were introduced on admission for rapidly progressive glomerulonephritis. A renal biopsy performed on Day 2 revealed only ATI. Therefore, steroids were discontinued on Day 5, after which renal function recovered spontaneously, and urinalysis abnormalities disappeared. Renal function remained normal during follow-up. We report a case of AKI with severe renal dysfunction after COVID-19 vaccination, wherein renal biopsy effectively determined the disease status (ATI), which did not require immunosuppressive treatment.

## 1. Introduction

The coronavirus disease 2019 (COVID-19) pandemic has resulted in a sudden and significant increase in pneumonia hospitalizations. It often leads to multiple organ failure and follows a severe course. COVID-19 multiorgan failure includes reduced function of the heart, brain, lungs, liver, kidneys, and coagulation system. The overall COVID-19 hospital mortality rate is approximately 15% to 20%; however, it approaches 40% in patients requiring treatment in the intensive care unit. Hospital mortality rates increase with age, ranging from 35% in patients aged 70–79 years to more than 60% in those aged 80–89 years. Patients infected with COVID-19 presenting with a severe turnaround, such as sepsis, are more likely to suffer from long-term mental, physical, and cognitive dysfunction with other sequelae [1].

Therefore, COVID-19 vaccination is crucial in reducing disease severity and mortality rates. The three vaccines approved by the U.S. Food and Drug Administration, including the Pfizer-BioNTech, Moderna, and Janssen/Johnson & Johnson vaccines, have been shown to be safe and effective in preventing severe cases of COVID-19. In a randomized controlled trial conducted in multiple countries, the pooled results of an interim analysis indicated the safety and efficacy of Oxford AstraZeneca’s chimpanzee adenovirus vector vaccine ChAdOx1 nCoV-19 (AZD1222) against COVID-19 in adults aged 18 years and older [2]. The common adverse events following COVID-19 vaccination include mild to moderate tenderness at the injection site, fever, fatigue, body aches, and headache. However, acute kidney injury (AKI) after COVID-19 vaccination is a rare adverse reaction that has not been reported in previous trials [3,4,5]. Conversely, rare conditions requiring treatment with immunosuppressive agents, such as acute glomerulonephritis, have been reported as a cause of AKI after vaccination. COVID-19 vaccines reportedly result in adverse events in the form of renal diseases, such as minimal change disease [6], IgA nephropathy [7], vasculitis [8,9], acute tubulointerstitial nephritis [10], and thrombotic microangiopathy (TMA) [11]. Most cases require treatment with immunosuppressive drugs and often result in AKI. However, no case of AKI pathology with acute tubular injury (ATI) alone after receiving a COVID-19 vaccine has been reported to date. Herein, we present a case in which the AKI pathology was ATI only and demonstrate the importance of excluding patients with diseases that require treatment with immunosuppressive drugs, such as vasculitis and nephritis. Conservative treatment without immunosuppressive drugs can improve the patient’s condition when ATI is the sole cause of the AKI.

## 2. Case Report

A 54-year-old man with type 2 diabetes visited a local physician. He was highly obese with a body mass index of 36 kg/m^2^. He was treated with metformin and insulin, and he was not prescribed any other regular medication. No other underlying diseases were observed. Diabetic retinopathy, urinary protein, and occult blood were absent. Serum creatinine (sCr) values remained in the range of 1.0–1.3 mg/dL (estimated glomerular filtration rate, 46–62 mL/min/1.73 m^2^). He had no family history of renal diseases. He had received three COVID-19 vaccines at four-month intervals. Two of the vaccines were from Pfizer-BioNTech and one was from Moderna. Adverse reactions to the last three vaccinations were not serious, including joint pain and transient fever. Five months after receiving the third vaccine, he received a fourth COVID-19 vaccine. There was no history of COVID-19 infection. Three days after receiving the fourth COVID-19 vaccine (Pfizer-BioNTech), he became fatigued and visited a local doctor four days later. The sCr level was 4.72 mg/dL, and he was given supplemental fluids for dehydration. And, metformin was then discontinued.

The next day, the sCr level was further elevated and the patient was referred to our hospital, where he was urgently hospitalized. On admission, he was conscious and oriented, with blood pressure of 145/92 mm Hg, heart rate of 80 beats/min, and body temperature of 36.2 °C with no abnormalities noted in his vital signs. A physical examination revealed no remarkable findings. Although fatigue was noted, typical symptoms of vasculitis, such as skin rash and abnormal sensation in the extremities, were not observed. The blood test results showed an elevated white blood cell count (9600 cells/μL), normal hemoglobin level (14.7 g/dL), renal dysfunction (sCr, 7.09 mg/dL), elevated inflammatory response (C-reactive protein, 1.15 mg/dL), and poor glycemic control (HbA1c 7.8%). Anti-neutrophil cytoplasmic, anti-nuclear, and anti-glomerular basement membrane antibodies were absent. Lactic acidosis was not observed. The urinary N-acetyl-β-D-glucosaminidase level was normal (50 μg/L; reference value, <289 μg/L). However, the N-acetylglucosaminidase level was mildly elevated (37.2 IU/L; reference value, 0.7–11.2 IU/L). Glomerular hematuria was present, and urinary protein excretion was approximately 1 g/day. Computed tomography revealed no renal atrophy or hydronephrosis. Chronic renal failure and post-renal failure were ruled out.

There was no obvious infectious focus observed on imaging, and bacteriological studies were negative. Therefore, we ruled out infectious causes of AKI. Since the previous physician had administered supplemental fluids and there was no evidence of hypotension, he excluded prerenal AKI due to intravascular dehydration from the differential diagnoses. Thus, renal AKI was suspected, because of the presence of new urinary protein and glomerular occult blood. Rapidly progressive glomerulonephritis (RPGN) from vasculitis was suspected, and immediate treatment with immunosuppressive drugs was deemed necessary. Therefore, methylprednisolone pulse therapy (500 mg/day) was administered for three days from the day of admission, followed by prednisone (60 mg/day). To determine the cause of the AKI, a renal biopsy was performed on the second day he was hospitalized.

Twenty-two glomeruli were collected, two of which had global sclerosis. No abnormal findings were observed in the other glomeruli. The tubules showed vacuolar degeneration (Figure 1a,b) with mild lymphocyte infiltration (Figure 1c). Interstitial fibrosis was unremarkable in the interstitium and tubules (Figure 1d). Immunoglobulin (Ig) G, IgA, C1q, and C3 assay results were negative for the glomerulus, implying no glomerular involvement. Pathologically, the findings indicated ATI. Pathological examination ruled out acute glomerulonephritis, interstitial nephritis, vasculitis, and TMA. Additionally, there were no obvious findings of diabetic nephropathy. Therefore, a diagnosis of ATI after COVID-19 vaccination was made. Prednisone was discontinued the day after the pathology results became known (prednisolone was administered for five days). The patient’s sCr level improved to 1.14 mg/dL 14 days after vaccination (at discharge), then to 1.00 mg/dL 19 days after vaccination. The sCr levels remained unchanged without worsening 3 months after discharge from the hospital. A chart of the treatment process is shown in Figure 2.

## 3. Discussion

The COVID-19 vaccines are important for preventing lethal outcomes caused by COVID-19. Additionally, the COVID-19 vaccine is generally considered safe. Minor side effects commonly observed after vaccination include pain at the injection site, fever, fatigue, and headache. AKI after vaccination is considered a rare adverse event and has not been reported in clinical trials. In a previous real-world analysis, kidney and urinary tract diseases accounted for only 0.46% of adverse events [12].

As mentioned earlier, reports on the cause of AKI after vaccination include mostly cases in which immunologic mechanisms are considered [6,7,8,9,10,11]. Additionally, many of these conditions require treatment with immunosuppressive agents. Glomerulonephritis caused by the COVID-19 vaccine has been reported to be frequently associated with AKI [13]. In addition, some of these patients reportedly progress to having end-stage kidney disease (ESKD) and require renal replacement therapy. Therefore, it is important to evaluate the presence of conditions such as acute glomerulonephritis as a cause of AKI following COVID-19 vaccination. These conditions are likely to require immunosuppressive drugs. AKI is associated with a poor prognosis due to the development of chronic kidney disease (CKD) and ESKD, which leads to higher mortality rates [14,15]. In this case, the pathology was ATI only, and immunosuppressive drugs were not indicated. Renal ischemia is thought to be involved in ATI pathogenesis [16], and supportive care is the mainstay of ATI treatment. On the other hand, high doses of methylprednisolone may have inhibited the progression of AKI. Furthermore, the main cause of AKI after COVID-19 vaccination is reportedly volume depletion (40.78%) followed by sepsis (11.74%), and the Pfizer-BNT vaccine has the highest incidence of AKI among all vaccines, requiring special consideration [17]. It has been reported that 42.1% of patients with sepsis have concurrent AKI, and septic AKI is associated with AKI of greater severity and higher mortality [18]. Therefore, we believe that it is important to be aware of the coexistence of infectious diseases and post-vaccination AKI. In this case, acute glomerulonephritis and sepsis were ruled out as causes of AKI. Although the patient did not show signs of hypotension or intravascular dehydration after admission, he may have been hypotensive prior to admission or the COVID-19 vaccine could have caused local renal perfusion defects. Previous reports of AKI after vaccination have demonstrated conditions requiring immunosuppressive agents, such as acute glomerulonephritis, as evidenced by the renal biopsy results. Therefore, in cases in which infection, obvious hypotension, or intravascular dehydration is not proven as a cause of AKI after COVID-19 vaccination, renal biopsy may be important to elucidate the pathogenesis of the disease. In the present case, prednisone was discontinued immediately after the diagnosis of ATI, after which renal function continued to improve. The clinical course corresponded to that of ATI. Mortality rates among patients who developed AKI after COVID-19 vaccination are reportedly high: 19.78% in the Pfizer BNT group, 17.78% in the Moderna group, and 12.36% in the JANSSEN group [17]. Therefore, despite conservatively treating patients with AKI, inpatient management may be considered in order to monitor them for the development of severe complications, such as renal failure requiring renal replacement therapy. Furthermore, AKI is more likely to occur in older adults (median age of 63 years), men (57%), those with comorbidities (for instance, diabetes 28%), and CKD (26%) [19]. Our patient met these characteristics; therefore, the patient was considered to have a high risk of developing AKI after vaccination. A previous publication reported a case of multiorgan failure associated with AKI after COVID-19 vaccination, wherein the renal biopsy results proved ATI [20]. The cause of the ATI was thought to be a combination of sepsis, hypotension, and non-steroidal anti-inflammatory drug (NSAID) administration. Although we ruled out ATI due to sepsis, hypotension, or the use of NSAIDs in our case, we should consider the possibility of severe multiorgan failure after vaccination.

RPGN from antineutrophil cytoplasmic antibody-related vasculitis and other causes has poor renal prognosis without early intervention with immunosuppressive drugs [21]. Therefore, early detection, diagnosis, and treatment of suspected cases of RPGN are important. At the time of initial diagnosis, it was difficult to differentiate ATI from RPGN because of vasculitis or other causes based on only clinical findings and laboratory tests. In addition, according to previous reports, the time lag between COVID-19 vaccination and the onset of RPGN varies from 3 h to 4 weeks, making it difficult to determine whether glomerular injury requiring immunosuppressive treatment occurs only within the time frame between vaccination and the onset of AKI [22]. Therefore, in cases of rapidly progressing renal dysfunction, empirical treatment with immunosuppressive agents may be unavoidable without waiting for a definitive diagnosis by renal biopsy. In this case, prednisone was administered before the results of the renal biopsy were available, considering the rapid progression of renal dysfunction in RPGN, as is seen in anti-glomerular basement membrane disease (GBM) [23]. The patient was diagnosed with a condition that did not require treatment with immunosuppressive drugs, leading to an early discontinuation of prednisone; therefore, no prednisone-associated adverse events occurred. Although rare, ATI caused by Pfizer-BioNTech COVID-19 vaccination has been reported, and this case may have been similar [17].

## 4. Conclusions

In conclusion, we reported a case in which a COVID-19 vaccine might have caused AKI with ATI as the sole pathology, which may not require immunosuppressive drugs. On the other hand, acute glomerulonephritis caused by a COVID-19 vaccine requires immunosuppressive drugs, without which the disease may have a severe course, including progression to ESKD. Therefore, in cases of AKI after COVID-19 vaccination with an unclear cause, such as sepsis, performing an aggressive renal biopsy should be considered for differential diagnosis. To the best of our knowledge, this is the first reported case of pathologically proven ATI after a COVID-19 vaccine. Although mechanisms and cases need to be accumulated, physicians need to be aware that ATI might occur after COVID-19 vaccination. 

## Figures and Tables

**Figure 1 vaccines-11-00464-f001:**
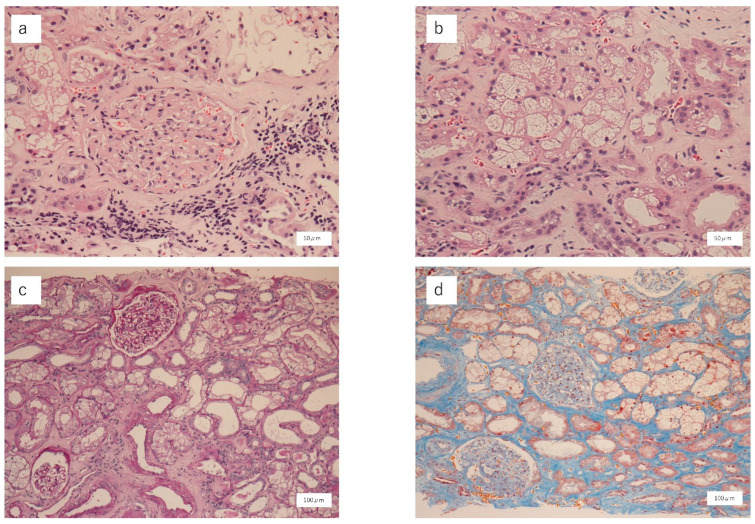
Renal biopsy findings. (**a**,**b**) Hematoxylin and eosin staining showing two glomeruli with global sclerosis. The other glomeruli are normal. The tubules are marked by vacuolar degeneration. (**c**) Periodic acid–Schiff staining showing mild lymphocyte infiltration in the interstitium and tubules. (**d**) Elastica–Masson staining showing unremarkable interstitial fibrosis in the interstitium and tubules.

**Figure 2 vaccines-11-00464-f002:**
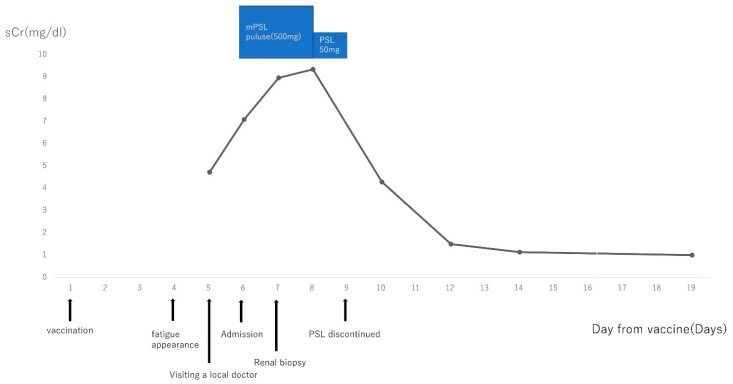
A chart of the treatment process. Rapidly progressive glomerulonephritis was suspected, and an initial treatment with methylprednisolone pulse therapy was given. After the renal biopsy results revealed acute tubular injury as the sole pathology, steroid treatment was discontinued. Eventually, renal function began to improve, and the serum creatinine level returned to baseline. sCr, serum creatinine; mPSL, methylprednisolone; PSL, prednisone.

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
