# Peer review of "Renal Biopsy Diagnosis of Acute Tubular Injury after Pfizer-BioNTech COVID-19 Vaccination: A Case Report"

_vaccines, 2023, doi:10.3390/vaccines11020464_

Round 1

Reviewer 1 Report

Estimated Authors of the case report "Renal Biopsy Diagnosis of Acute Tubular Injury After Pfizer-BioNTech COVID-19 Vaccination: A Case Report"

I've read with great interest your article. This is a case report well documented and well written but, unfortunately, it is also affected by several shortcomings from a scientific point of view. The 54-y.o. individual you're reporting on was affected by AKI by Acute Tubular Injury. ATI is a well pattern characterized by specific histological changes of the tubules. After loss of brush border and nonisometric cytoplasmic vacuolization, tubular cells undergo cytoplasmic fragmentation or detach entirely into the lumen, producing obstructive casts. In this case, the pathological findings were considered compatible with the definition of ATI, and while this is not particularly problematic, the following diagnosis of COVID-19 vaccine-related ATI it is.

In fact, a rapid literature research on the topic COVID-19 vaccine and acute tubular injury produces very few evidence, mostly rather associated with ASTRA ZENECA and even SARS-CoV-2 infection rather than mRNA vaccine. In other words, we're dealing with a temporal correlation interpreted as a causation. 

FRom the point of view of the present reviewer, while the topic and the reporting are appropriate, interesting, and consistent with the aims and quality of Vaccines, Authors should forcibly reshape discussion (at least partially) in a more cautious way, stressing how the case here reported may be only a fortuitous consequence.

Author Response

We would like to express our gratitude for your insightful comments, which have allowed us to revise the paper and make significant improvements.

  1. In fact, a rapid literature research on the topic COVID-19 vaccine and acute tubular injury produces very few evidence, mostly rather associated with ASTRA ZENECA and even SARS-CoV-2 infection rather than mRNA vaccine. In other words, we're dealing with a temporal correlation interpreted as a causation.

Response: Thank you for raising concern. As you pointed out, acute tubular necrosis (ATN) with mRNA vaccines is not common. However, although less frequently, ATN has also been reported with mRNA vaccines such as Pfizer-BNT and Moderna [1]. Also, the time of onset after vaccination was around 10 days, and we believe that ATN due to Pfizer-BNT vaccination is a possibility in this case. As you pointed out, we cannot make a determination, so we have modified the text to better convey this point. We have revised the red highlighted text on page 5, lines 189-190.

  1. Huiting Luo, Xiaolin Li, Qidong Ren, et al. Acute kidney injury after COVID-19 vaccines: a real-world study. Ren Fail. 2022 Dec;44(1):958-965

  1. From the point of view of the present reviewer, while the topic and the reporting are appropriate, interesting, and consistent with the aims and quality of Vaccines, Authors should forcibly reshape discussion (at least partially) in a more cautious way, stressing how the case here reported may be only a fortuitous consequence.

Response: Thank you for your valuable suggestion. As you pointed out, it is difficult to draw a definitive conclusion that the ATI is due to vaccination, and we have revised the wording of the text to better convey this.

Reviewer 2 Report

The manuscript is well-written and it presents an interesting case report. It strictly links the AKI with the vaccine. Still, this must be seen as a hypothesis. 

The discussion is lengthy in words and should be shortened. It should be mentioned that the high dose of prednisone may be had an effect to stop the ongoing process which had initiated the ATI.  This should be presented as a hypothesis.

When was the metformin treatment stopped? This should be mentioned in the Case report.

Minor point: the word vessels in line 182 must be replaced by vasculitis.

Author Response

We would like to express our gratitude for your insightful comments, which have allowed us to revise the paper and make significant improvements.

  1. The manuscript is well-written and it presents an interesting case report. It strictly links the AKI with the vaccine. Still, this must be seen as a hypothesis.

Response: Thank you for your valuable suggestion. As you pointed out, it is difficult to draw a definitive conclusion that the ATI is due to vaccination, and we have revised the wording accordingly.

  1. The discussion is lengthy in words and should be shortened.

Response: We appreciate your appropriate this appropriate helpful recommendation. Some of the text in Discussion has been removed and made more concise. Specifically, we have removed 160 words to shorten the Discussion section.

  1. It should be mentioned that the high dose of prednisone may be had an effect to stop the ongoing process which had initiated the ATI. This should be presented as a hypothesis.

Response: Thank you for this beneficial comment. We have revised the red highlighted text on page 4, line 148-149.

  1. When was the metformin treatment stopped? This should be mentioned in the Case report.

Response: Thank you for pointing this out. We have added a description of metformin discontinuation in the red highlighted text on page 2, line 73. In addition, lactic acidosis was not observed in the red highlighted text on page 2, line 85.

  1. Minor point: the word vessels in line 182 must be replaced by vasculitis.

Response: We appreciate your astute observation. We have corrected this error on page 5, line 176.

Round 2

Reviewer 1 Report

Authors have done a decent work in promoting a more cautious description of this case report.

I'm endorsing the eventual acceptance of the paper.

Reviewer 2 Report

The authors have responded to suggestions accordingly.